# COMBINING MODEL-BASED AND MODEL-FREE RL VIA MULTI-STEP CONTROL VARIATES

## ABSTRACT

Model-free deep reinforcement learning algorithms are able to successfully solve a wide range of continuous control tasks, but typically require many on-policy samples to achieve good performance. Model-based RL algorithms are sample-efficient on the other hand, while learning accurate global models of complex dynamic environments has turned out to be tricky in practice, which leads to the unsatisfactory performance of the learned policies. In this work, we combine the sample-efficiency of model-based algorithms and the accuracy of model-free algorithms. We leverage multi-step neural network based predictive models by embedding real trajectories into imaginary rollouts of the model, and use the imaginary cumulative rewards as control variates for model-free algorithms. In this way, we achieved the strengths of both sides and derived an estimator which is not only sample-efficient, but also unbiased and of very low variance. We present our evaluation on the MuJoCo and OpenAI Gym benchmarks.

## 1 INTRODUCTION

Reinforcement learning algorithms are usually divided as two broad classes: model-based algorithms which try to learn forward dynamics models first (and derive a planning-based policy from it), and model-free algorithms which do not explicitly train any dynamics models but instead directly learn a policy. Model-free algorithms have been proven effective in learning sophisticated controllers in a wide range of applications, ranging from playing video games from raw pixels Mnih et al. (2013); Oh et al. (2016) to solving locomotion and robotic tasks Schulman et al. (2015). Model-based algorithms on the other hand are widely used in robot learning and continuous control tasks Deisenroth & Rasmussen (2011).

Both approaches have their strengths as well as limitations. Although model-free RL algorithms can achieve very good performance, they usually require ten thousands of samples for each iteration (Schulman et al., 2015), which is a very high sample intensity. A fundamental advantage of model-based RL is that knowledge about the environment can be acquired in an unsupervised setting, even in trajectories where no rewards are available. While model-based algorithms are usually considered as more sample-efficient (Deisenroth et al., 2013), currently the most successful model-based algorithms rely on simple functional approximators Lioutikov et al. (2014) to learn local models over-fitting a few samples, usually one mini-batch. Thus ideas that rely on a global neural network model for planning often do not perform very well (Mishra et al., 2017; Gu et al., 2016b) due to the bias of the learned models and the limitations of the planning algorithms, e.g. MCTS (Chaslot, 2010). As a result, although theoretically global models should be more sample efficient because models can be trained on off-policy data, they are seldom used in real problems.

In this paper, we attempt to combine the benefits of model-based and model-free reinforcement learning and reduce the drawbacks on both sides. Our algorithm can generally be viewed as using a multi-step forward model as a control variate (MSCV) for model-free RL algorithms. It works as follows: given a batch of on-policy trajectory samples, instead of directly performing on-policy gradient update based on the return, we first embed a portion of the trajectory into the dynamics model, yielding an imaginary trajectory which is usually very close to the real trajectory thus we call it a trajectory "embedding" to the dynamics model. From the imaginary trajectory, we can get low variance gradients via the reparameterization trick Kingma & Welling (2013) and backpropagation-through-time (BPTT) Werbos (1990). However, such gradients are in general biased. The funda-

mental reason for this is that in real circumstances, the global dynamics is highly complicated while the model is usually biased. Thus afterwards, we use the return from the real trajectory to correct the model-based gradient. This way, we effectively combine the strength of both model-based and model-free RL while avoiding their shortcomings. We evaluated MSCV on OpenAI Gym Brockman et al. (2016) and MuJoCo Todorov et al. (2012) benchmarks.

The contributions of our work are the following: (1) We show how to train efficient multi-step value prediction models by utilizing techniques such as replay memory and importance sampling on off-policy data. (2) We demonstrate that the model-based path derivative methods can be combined with model-free RL algorithms to produce low variance, unbiased, and sample efficient policy gradient estimates. (3) We show that techniques such as latent space trajectory embedding and dynamic unfolding can significantly boost the performance of the model based control variates.

## 2    RELATED WORKS

Model-based and model-free RL are two broad classes of reinforcement learning algorithms with different weakness and strength. Thus no doubt there are many works trying to combine the strength of both sides and avoid their weakness. (Nagabandi et al., 2017) tries to train a neural network based global model and use model predictive control to initialize the policy for model free fine tuning. In this way, their algorithm can be significantly more sample-efficient than purely model free approaches. (Chebotar et al., 2017) combines local model based algorithms such as LQR to a model-free framework called path integral policy improvement. Local models such as LQR are sample efficient but they need to refit simple local models for each iteration, so they cannot work with high dimensional state spaces such as images.

QProp (Gu et al., 2016a) proposed to use a critic trained over the amortized deterministic policy using DDPG (Lillicrap et al., 2015) as a control variate. While this has brought good performance gain in terms of sample complexity, our new control variate provides several attractive properties over it. First, the Q function $Q(s, a)$ fitted off-policy using DDPG is usually very different from the real on-policy expected return $Q^\mu(s, a)$ (because the trajectory in the replay buffer are generated by different policies, they have different value estimates). As we've seen in our experiments, sometimes the gap between these two values is of orders of magnitude. However, in our algorithm, since their is no inconsistency of the objectives, our control variate is usually much more accurate than QProp. See e.g. Section 5. Second, Our control variate takes into consideration not only the current policy step, but also several future steps. This approach have brought two main gains to our algorithm: (1) our control variate can be fitted much closer to the actual return, resulting in less variance. (2) we can use backpropagation through time to optimize multiple steps in the imaginary trajectory jointly, this makes our algorithm be able to look further into the future. This is very useful when the reward cannot be observed very often.

Our algorithm is also closely related to stochastic value gradient (Heess et al., 2015). In fact, if the control variate choose to unroll our multi-step model for one step, then the resulting algorithm is equivalent to use SVG(1) algorithm in (Heess et al., 2015) as a control variate for a model-free algorithm. However, our algorithm makes use of multi-step prediction and dynamic rollout which SVG did not take advantage of.

## 3    PRELIMINARIES

Reinforcement learning aims at learning a policy that maximizes the sum of future rewards (Nagabandi et al., 2017). Formally, in an MDP environment (Thie, 1983) $(S, A, f, r)$, suppose that at time $t$ the agent is in state $\mathbf{s}_t \in S$ and it executes one action $\mathbf{a}_t$ sampled according to its policy $\pi_\theta(\mathbf{a}_t | \mathbf{s}_t)$, then receives a reward $r(\mathbf{s}_t, \mathbf{a}_t)$ from the environment. Following this, the environment makes a transition to state $\mathbf{s}_{t+1}$ according to an unknown dynamics function $f(\mathbf{s}_t, \mathbf{a}_t, \eta)$ where $\eta$ is a random noise variable sampled from a fixed distribution, for example $\eta \sim N(0, 1)$. The trajectory $\{s_i, a_i, r_i\}_{i=1}^N$ is the resulting sequence of state-action-reward records produced by the policy and the environment. The goal is to maximize the discounted total sum of reward $\sum_{t'=t}^{+\infty} \gamma^{t'-t} r(\mathbf{s}_{t'}, \mathbf{a}_{t'})$, where $\gamma$ is a discounting factor that prioritizes near-term rewards. In MSCV, to select the best actions, we train a stochastic policy $\mathbf{a} = \mu(\mathbf{s}, \epsilon) \sim \pi_\theta(\mathbf{a} | \mathbf{s})$, where $\epsilon \sim N(0, 1)$ is another random noise variable.

In model-based reinforcement learning, a dynamics model is learned as an approximation of the real unknown dynamics $f$. In MSCV, we are learning a multi-step dynamics model $\hat{f}$ predicting the next state $\mathbf{s}_{k+1} = \hat{f}(\mathbf{s}_1, \mathbf{a}_1, \eta_1 \cdots \mathbf{a}_k, \eta_k)$. $\hat{f}$ is modeled as a RNN so that it can be unrolled for an arbitrary number of time steps $k$. We also need to fit a reward predictor $\hat{r}(\mathbf{s}_t, \mathbf{a}_t)$ (Otherwise we can also assume the reward function $r$ is available to us.). Besides these two state and reward predictive models, we also learn an on-policy value function $\hat{V}^\pi(\mathbf{s})$ from off-policy data, which can be viewed as an approximation of the real expected future return $V^\pi(s)$ of state $\mathbf{s}$.

## 4 MULTI-STEP MODELS AS CONTROL VARIATES

Assume that we have a multi-step model as specified above. Given a real trajectory $\mathcal{T} = \{s_i, a_i, r_i, \eta_i, \epsilon_i\}_{i=t}^N$ collected using the current policy $\pi_\theta$, we define the empirical $k$-step return as $\hat{Q}_k^\pi(s_t, a_t) = r_t + \cdots + \gamma^{k-1} r_{t+k-1} + \gamma^k \hat{V}^\pi(s_{t+k})$.

Given the sub-trajectory $T_k = \{s_i, a_i, r_i, \eta_i, \epsilon_i\}_{i=t}^{t+k}$, we can "embed" $T_k$ to an imaginary trajectory $E(\mathcal{T}) = \{s_i', a_i', r_i', \eta_i, \epsilon_i\}_{i=t}^{t+k}$ where $E(\mathcal{T})$ satisfies three conditions: (1) $s_t' = s_t$, (2) $\{s_{i+1}' = \hat{f}(s_i', a_i', \eta_i)\}_{i=t}^{t+k}$ (3) $\{a_i' = \mu(s_i', \epsilon_i)\}_{i=t}^{t+k}$. One can easily verify $\hat{f}$ and $\mu_\theta$'s effectiveness by seeing that when $\hat{f} = f$, the imaginary trajectory $E[\mathcal{T}]$ perfectly matches the real trajectory $\mathcal{T}$.

In the following, for notational simplicity we assume the unknown dynamics $f$ and the model $\hat{f}$ are deterministic. The stochastic cases are similar except that one need an extra inference network to infer the latent variables of a given generation. We can represent the $k$-step imaginary empirical return as:

$$\bar{Q}(s_t, \mu_\theta(s_t, \epsilon_t), \epsilon_{t+1} \cdots \epsilon_{t+k-1}) = \bar{Q}(s_t, a_t' \cdots a_{t+k-1}') \tag{1}$$
$$= \hat{r}(s_t', a_t') + \cdots + \gamma^{k-1} \hat{r}(s_{t+k-1}', a_{t+k-1}') + \gamma^k \hat{V}^\pi(s_{t+k}'),$$

where $a' = \mu_\theta(s', \epsilon)$. Assume all $\{\epsilon_i\}_{i=k}^{t+k}$ are sampled from fixed prior distribution $P = N(0, 1)$, we have the following policy gradient formula:

$$\nabla J(\theta) = \mathbb{E}_{\rho^\pi, P}[\nabla_\theta \log \pi(a_t|s_t) \hat{Q}_k^\pi(s_t, a_t)] \tag{2}$$
$$= \mathbb{E}_{\rho^\pi, P}[\nabla_\theta \log \pi(a_t|s_t)[\hat{Q}_k^\pi(s_t, a_t) - \bar{Q}(s_t, a_t, \epsilon_{t+1} \cdots \epsilon_{t+k-1})]] \tag{3}$$
$$+ \mathbb{E}_{\rho^\pi, P}[\nabla_\theta \log \pi(a_t|s_t) \bar{Q}(s_t, a_t, \epsilon_{t+1} \cdots \epsilon_{t+k-1})] \tag{4}$$

We assume in the above equation that the imaginary rollout to compute $\bar{Q}$ is an embedding of the real rollout to compute $\hat{Q}$. The second term can be expressed via the reparameterization trick:

$$\mathbb{E}_{\rho^\pi, P}[\nabla_\theta \log \pi(a_t|s_t) \bar{Q}(s_t, a_t, \epsilon_{t+1} \cdots \epsilon_{t+k-1})] = \int_a \nabla_\theta^t \mathbb{E}_{\rho^\pi, P}[\pi_\theta(a|s_t) \bar{Q}(s_t, a, \epsilon_{t+1} \cdots \epsilon_{t+k-1})] da$$
$$= \nabla_\theta^t \int_a \mathbb{E}_{\rho^\pi, P}[\pi_\theta(a|s_t) \bar{Q}(s_t, a, \epsilon_{t+1} \cdots \epsilon_{t+k-1})] da$$
$$= \nabla_\theta^t \mathbb{E}_{\rho^\pi, P}[\bar{Q}(s_t, \mu_\theta(s_t, \epsilon_t), \epsilon_{t+1} \cdots \epsilon_{t+k-1})], \tag{5}$$

where the operator $\nabla_\theta^t$ stands for the partial derivative with respect to the $\theta$ of time step $t$ only, namely we treat policy $\pi_\theta$ occurring before and after time step $t$ as fixed. This can be easily done by detaching the node from the computational graph with any modern deep learning framework exploiting automatic differentiation. Through this way, the variance of the second term $G_2$ can be reduced significantly since we cut off its dependency among former and latter timesteps. The algorithm we use for calculating the partial derivative of $G_2$ is backpropagation through time of RNN.

There are several interesting properties of the derived gradient estimator. First, imaging the extreme cases in which $\hat{f}, \hat{r}$ are perfect, then $G_1$ is always zero. On the other hand, $G_2$ is reparameterized as a differentiable function from a fixed random noise, whose gradient is of low variance. In practice, one can replace $k$-step cumulative return $\hat{Q}_k$ with the real return $\hat{Q}$ of the entire trajectory. In this case, the estimator will be unbiased at the expenses of a little more variance.

**Dynamic rollout.** There is a trade-off on the model rollout number $k$. If $k$ is too small, the model bias is small but the training procedure is not able to take advantage of the multi-step trajectory optimization and the accurate reward predictions. Moreover, more burden would be put on the accuracy

of $\hat{V}^\pi$. If $k$ is too large, the model bias will take over, and the imaginary trajectory will diverge with the real trajectory, but the bias in the value function is relieved because of the discounting. A proper $k$ will help us balance the biases from the forward model and the value function.

In our algorithm, we perform a dynamic model rollout, following the similar technique with QProp. Namely, let $\hat{A} = \hat{Q} - \hat{V}$ be the advantage estimation of the real trajectory, and $\bar{A} = \bar{A} - \hat{V}$ be the advantage estimation of the generated trajectory. We perform a short line search on $k$, such that $k$ is the largest number such that $Cov(\hat{A}, \bar{A}) > 0$.

---

**Algorithm 1** $k$-step Actor-Critic with Model-based Control Variate

---

$\mu(s, \epsilon)$**: Policy network parameterized by** $\theta_\mu$
$V(s)$**: Value network. parameterized by** $\theta_v$
$f(s, a)$**: Forward model, parameterized by** $\theta_f$
$T$**: time step horizon for TRPO update.**
$L$**: Maximum episode length for data collection.**
$k$**: number of time steps to evaluate return.**
$\mathcal{M}$**: the replay memory.**

    Initialize $\mu, V$ randomly, initialize $f$ to pretrained model, initialize environment states $S$.
    **for** $\mu$ not converge **do**
        Collect batch $\mathcal{B}$ of trajectories $\{s_{i,e}, \epsilon_{i,e}, a_{i,e} \sim \mu_\theta, r_{i,e}\}_{i=0,e=1}^{T-1,N}$ with $\{s_{0,e}\}$ from $S$.
        **if** episode ends or length $> L$ **then**
            reset $S$ to initial states.
        **end if**
        Add $\mathcal{B}$ to $\mathcal{M}$.
        **for** $n$ iterations **do**
            Sample batch $\mathcal{B}_f$ of trajectories of length $k$ from $\mathcal{M}$ and train model $f$ on $\mathcal{B}_f$.
            Sample batch $\mathcal{B}_v = \{s_i, a_i, r_i, s_i'\}_{i=1}^N$ of transitions from $\mathcal{M}$, train $V$ with the algorithm
    specified in section 5.
        **end for**
        $d\theta_\mu = 0$.
        **for** each $t = 0, 1, \cdots T - k$ **do**
            Take a batch of sub-trajectories $\mathcal{B}_t = \{s_{j,e}, a_{j,e}, \epsilon_{j,e}, r_{j,e}\}_{j=t,e=1}^{t+k-1,N}$ from $\mathcal{B}$
            Unroll the model for $k$ steps, get $\mathcal{B}_t' = \{s_{j,e}', a_{j,e}', \epsilon_{j,e}, r_{j,e}'\}_{j=t,e=1}^{t+k-1,N}$ such that $s_{t,e}' = s_{t,e}$
    and fix the latent variables $\epsilon$ as in $\mathcal{B}_t$.
            Calculate gradient $G_1 = \frac{1}{N} \sum_e [\nabla_\theta \log \pi(a_t|s_t)[\hat{Q}_k^\pi(s_t, a_t) - Q(s_t, a_t, \epsilon_{t+1} \cdots \epsilon_{t+k-1})]$.
            Estimate the gradient $G_2 = \nabla_\theta^t \mathbb{E}_{\rho^\pi, P} [\bar{Q}(s_t, \mu_\theta(s_t, \epsilon_t'), \epsilon_{t+1}' \cdots \epsilon_{t+k-1}')]$ by setting initial
    state to be $s_{t,e}$ and perform multiple random model roll-outs and backpropagate through time.
            Accumulate gradients: $d\theta_\mu = d\theta_\mu + G_1 + G_2$
        **end for**
        Update $\theta_\mu$ with $d\theta_\mu$ with TRPO.
    **end for**

---

## 5    OFF-POLICY LEARNING OF MULTI-STEP MODELS AND VALUE FUNCTIONS

**Dynamics model.** We now present how to off-policy learn the multi-step model and the value function. We use a replay buffer to train both of them. It is relatively straight forward to train the multi-step model $\hat{f}$ given a batch of length $k$ trajectories. We use similar techniques with (Nagabandi et al., 2017) to predict the difference between consecutive states. Therefore, we have $s_{t+1}' = \hat{f}(s_t, a) + s_t$, which gives rise to a esNet like structure. This is a suitable structure because it fits the prior of MDP that the environment need to know the information of this state to generate the next state. In this work, we assume the reward function $r$ is accessible. Then the overall objective is

$$s_{t+1}' - s_t)^2 + \alpha(r_t - r(s_t, s_{t+1}', a_t)^2$$

where $\alpha$ is the weight for reward loss. In the experiment, we use $\alpha = 10$.

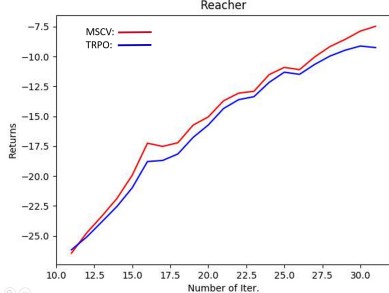

Figure 1: Reacher

Table 1: Iteration Required for Every Task

| Task | Threshold | MSCV | TRPO |
|---|---|---|---|
| Swimmer | 90 | **21** | 30 |
| Reacher | -7 | **34** | 42 |
| Walker | 2500 | **167** | 279 |
| Hopper | 2000 | **35** | 57 |

**Value function.** The value function $\hat{V}^\pi$ can be conveniently fitted off-policy. Different from DDPG, the gradient estimator for $\hat{V}^\pi$ is fitted using importance sampling, thus the objective is consistent. More precisely, if a transition $(s, a, r, s)$ sampled from the replay buffer is generated by policy $\pi_{\theta_0}(a|s)$, the current policy is $\pi_\theta(a|s)$, we compute an importance weight $w_t = \frac{\pi_\theta(a_t|s_t)}{\pi_{\theta_0}(a_t|s_t)}$, and we minimize the weighted bellman error: $w_t(\hat{V}(s, \theta) - y)^2$, where $y = r + \hat{V}'(s, \theta)$.

---

**Algorithm 2** Fitting Value Function $V^\pi$

---

**Given experience replay memory** $M$
**Given value function** $\hat{V}(\cdot, \theta)$**, outer loop time t**
    $\theta = \theta'$
    **for** $m = 0$ **do**
        Sample$(s^k, \mathbf{a}^k, r^k, s^{k+1})$ from $M(k < t)$.
        $y^m = r^k + \gamma\hat{V}(s^{k+1}; \theta')$
        $w = \frac{p(\mathbf{a}^k|s^k)}{p(\mathbf{a}^k|s^k)}$.
        $\Delta = \nabla_{\nu^{new}} \frac{w}{2}(y^m - \hat{V}(s^k; \theta))^2$.
        Apply gradient-based update to $\theta$ using $\Delta$.
        $\theta' = \sigma\theta' + (1 - \sigma)\theta$
    **end for**

---

## 6 EXPERIMENT

We evaluate MSCV on the MuJoCo and OpenAI Gym benchmarks against TRPO. We perform experiments on Reacher, Hopper, Swimmer and Walker tasks. The batch size is 5000 for the first three tasks and 25000 for Walker. For all experiments we set the number of steps $k$ to be 2. Although we tried other values of $k$, we found $k = 2$ is most effective. We set the discounting factor to be 0.995 and $\alpha = 10$ for all experiments.

### 6.1 MULTI-STEP DYNAMICS MODEL

In this section, we evaluate the dynamics model's capacity. We use the similar architecture with that in (Nagabandi et al., 2017) to predict the difference between the next states and current states.

Figure 3 shows our predicted rewards in Swimmer task. We randomly choose a starting state from a trajectory and roll out for 15 steps. Nevertheless, it is not enough to have accurate reward in our algorithm, because we would backpropogate through the trajectory as well. In Figure 2, we compare the L2-norm of the actual states and imaginary states from the dynamic model. We also plot the norm of difference between two states and the cosine similarity. The results show that the dynamics model is able to predict the future states and rewards, which proves the fundamental assumptions in our algorithm.

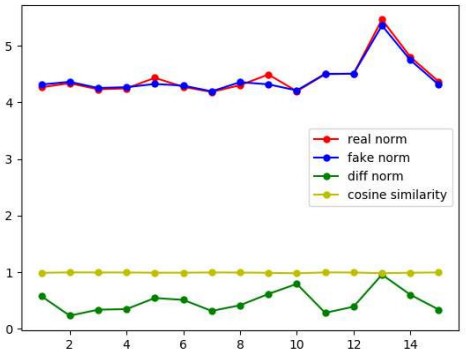

Figure 2: State Prediction

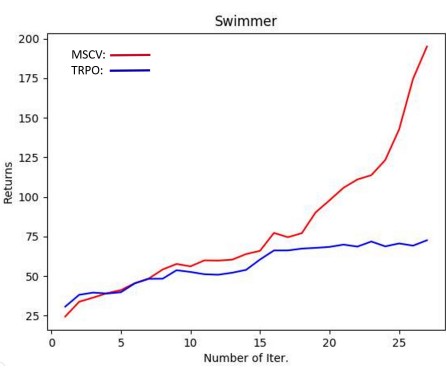

Figure 3: Reward Prediction

Figure 4: Swimmer

Figure 5: Walker

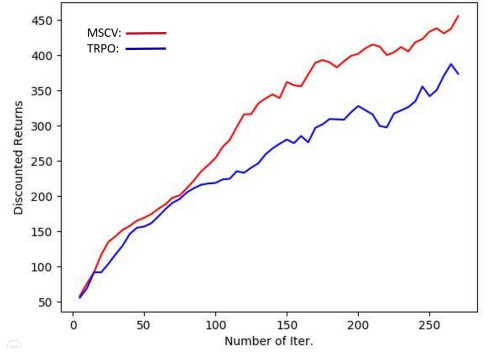

## 6.2 EVALUATION ACROSS DOMAINS

In this section, we present the comparison between MSCV and TRPO. For all tasks, both policy and value networks are two-layer neural networks with hidden size $[64, 64]$. We pre-train the dynamics model for each task using the technique described in section 5.

We summarize the results in Table 1. We set a threshold for each task and recorded the number of iteration needed in order to reach the threshold. Table 1 shows that MSCV outperformed TRPO in every task. The most significant improvement is in Walker, where the setting is more complex than the other three. It shows that by combining model-free and model-based algorithms, we can learn to improve the policy more efficiently in the complex dynamics.

Figure 4, 5, and 1 shows the improvement of policy after each iteration. It shows MSCV is able to converge faster and learn more efficiently than TRPO at each iteration. The improvement in Figure 1 is least significant, and we believe that it is because of the simple dynamic in this task that TRPO is already able to learn efficiently.

## 7 CONCLUSION

In this paper we presented the algorithm MSCV which combines the advantages of model-free and model-based reinforcement learning algorithms while alleviates the shortcomings of both sides. The resulting algorithm utilizes BPTT and multi-step models as a control variate for model free algorithms.

Our algorithm can be viewed as a multi-step model-based generalization of QProp, in which, instead of training a Q function using unstable techniques such as DDPG, we use off-policy data to stably train a multi-step dynamics model, a reward predictor, and an on-policy value function.

Our experiments on MuJoCo and OpenAI Gym benchmarks not only prove the footstone assumptions of MSCV, but show that MSCV outperforms the baseline method significantly in various circumstances.

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
