# OpenReview forum: "Combining Model-based and Model-free RL via Multi-step Control Variates"
_ICLR.cc/2018/Conference — Reject_

### Official Review · AnonReviewer2 · 2017-11-24
**Lacking clarity and significance.**

**Rating:** 5
**Confidence:** 4

**Review:**

The paper studies a combination of model-based and model-free RL. The idea is to train a forward predictive model which provides multi-step estimates to facilitate model-free policy learning.  Some parts of the paper lack clarity and the empirical results need improvement to support the claims (see details below).

Clarity
- The main idea of the proposed method is clear.
- Some notations and equations are broken. For example:
(1) The definition of \bar{A} in Section 4 is broken.
(2) The overall objective in Section 5 is broken.
(3) The computation of w in Algorithm 2 is problematic.
- Some details of the experiments/methods are confusing. For example:
(1) The step number k is dynamically determined by a short line search as in Section 4 ``Dynamic Rollout’’, but later in the experiments (Section 6) the value of k is set to be 2 uniformly.
(2) Only the policy and value networks specified. The forward models are not specified.
(3) In algorithm 1, what exact method is used in determining if \mu is converged or not?

Originality
The proposed method can be viewed as a multi-step version of the stochastic value gradient algorithm. An empirical comparison could be helpful but not provided.

The idea of the proposed method is related to the classic Dyna methods from Sutton. A discussion on the difference would be helpful.

Significance
- The paper could compare against other relevant baselines that combine model-based and model-free RL methods, such as SVG (stochastic value gradient).
- To make a fair comparison, the results in Table 1 should consider the amount of data used in pre-training the forward models. Current results in Table 1 only compare the amount of data in policy learning.
- Figure 3 is plotted for just one random starting state. The Figure could have been more informative if it was averaged over different starting states.  The same issue is found in Figure 2.  It would be helpful if the plots of other domains are provided.
- In Figure 2, even though the diff norm fluctuates, the cosine similarity remains almost constant. Does it suggest the cosine similarity is not effective in measuring the state similarity?
- Figure 1, 4 and 5 need confidence intervals or standard errors.

Pros:
- The research direction in combining model-based and model-free RL is interesting.
- The main idea of the proposed method is clear.

Cons:
- Parts of the paper are unclear and some details are missing.
- The paper needs more discussion and comparison to relevant baseline methods.
- The empirical results need improvement to support the paper’s claims.

---

### Official Review · AnonReviewer1 · 2017-11-26
**Interesting idea; empirical section could have been better**

**Rating:** 5
**Confidence:** 4

**Review:**

The main idea of the paper is to improve off-policy policy gradient estimates using control variates based on multi-step rollouts, and reduce the variance of those control variates using the reparameterization trick. This is laid out primarily in Equations 1-5, and seems like a nice idea, although I must admit I had some trouble following the maths in Equation 5. They include results showing that their method has better sample efficiency than TRPO (which their method also uses under the hood to update value function parameters).

My main issue with this paper is that the empirical section is a bit weak, for instance only one run seems to be shown for both methods, there is no mention of hyper-parameter selection, and the measure used for generating Table 1 seems pretty arbitrary to me (how were those thresholds chosen?). In addition, one thing I would have liked to get out of this paper is a better understanding of how much each component helps. This could have been done via empirical work, for instance:
- Explore the effect of the planning horizon, and implicitly compare to SVG(1), which as the authors point out is the same as their method with a horizon of 1.
- Show the effect of the reparameterization trick on estimator variance.
- Compare the bias and variance of TRPO estimates vs the proposed method.

---

### Official Review · AnonReviewer3 · 2017-11-28
**Nice idea, but paper feel half-finished**

**Rating:** 4
**Confidence:** 3

**Review:**

This paper presents a model-based approach to variance reduction in policy gradient methods.  The basic idea is to use a multi-step dynamics model as a "baseline" (more properly a control variate, as the terminology in the paper uses, but I think baselines are more familiar to the RL community) to reduce the variance of a policy gradient estimator, while remaining unbiased.  The authors also discuss how to best learn the type of multi-step dynamics that are well-suited to this problem (essentially, using off-policy data via importance weighting), and they demonstrate the effectiveness of the approach on four continuous control tasks.

This paper presents a nice idea, and I'm sure that with some polish it will become a very nice conference submission. But right now (at least as of the version I'm reviewing), the paper reads as being half-finished.  Several terms are introduced without being properly defined, and one of the key formalisms presented in the paper (the idea of "embedding" an "imaginary trajectory" remains completely opaque to me.  Further, the paper seems to simply leave out some portions: the introduction claims that one of the contributions is "we show that techniques such as latent space trajectory embedding and dynamic unfolding can significantly boost the performance of the model based control variates," but I see literally no section that hints at anything like this (no mention of "dynamic unfolding" or "latent space trajectory embedding" ever occurs later in the paper).

In a bit more detail, the key idea of the paper, at least to the extent that I understood it, was that the authors are able to introduce a model-based variance-reduction baseline into the policy gradient term.  But because (unlike traditional baselines) introducing it alone would affect the actual estimate, they actually just add and subtract this term, and separate out the two terms in the policy gradient: the new policy gradient like term will be much smaller, and the other term can be computed with less variance using model-based methods and the reparameterization trick.  But beyond this, and despite fairly reasonable familiarity with the subject, I simply don't understand other elements that the paper is talking about.

The paper frequently refers to "embedding" "imaginary trajectories" into the dynamics model, and I still have no idea what this is actually referring to (the definition at the start of section 4 is completely opaque to me).  I also don't really understand why something like this would be needed given the understanding above, but it's likely I'm just missing something here.  But I also feel that in this case, it borders on being an issue with the paper itself, as I think this idea needs to be described much more clearly if it is central to the underlying paper.

Finally, although I do think the extent of the algorithm that I could follow is interesting, the second issue with the paper is that the results are fairly weak as they stand currently.  The improvement over TRPO is quite minor in most of the evaluated domains (other than possibly in the swimmer task), even with substantial added complexity to the approach.  And the experiments are described with very little detail or discussion about the experimental setup.

Nor are either of these issues simply due to space constraints: the paper is 2 pages under the soft ICLR limit, with no appendix.  Not that there is anything wrong with short papers, but in this case both the clarity of presentation and details are lacking.  My honest impression is simply that this is still work in progress and that the write up was done rather hastily.  I think it will eventually become a good paper, but it is not ready yet.

---

### Public Comment · (anonymous) · 2017-12-03
**Result in more challenging domain**

1) It would be very interesting to see if any improvements in sample efficiency can been seen in not so toy and more high dimensional domain. Starting at least from humanoid walking task. It's not clear at the moment if there are any benefits for really high-dimensional challenging tasks from the proposed algorithm.

2) Also why a baseline for comparison is only TRPO? Not more sample efficient PPO, the same Q-prop mentioned in paper, DDPG is also a good candidate for baseline. Looks like there won't be any advantage in sample efficiency compare to these baselines. TRPO is the most convenient choice for comparison.

---

### Public Comment · ~Tong_Che1 · 2018-01-06
**To all reviewers...**

Thank you very much for your reviews...
We acknowledge that the experiments section in the current version of this paper is not strong enough.
As all the authors agreed,  that we should submit a revised version of this paper to a later venue, adding more experimental numbers.

To reply your questions:

Review 3: Let me briefly explain the terminology here. Latent space trajectory embedding means that, given a real-world trajectory which is generated by the environment and the policy, we can "embed" the trajectory into an imaginary trajectory that is generated by the model and the policy. We keep the latent variables of the policy fixed so that if the model perfectly matches the environment, the imaginary trajectory perfectly matches the real trajectory, so this is what we called "trajectory embedding" in the paper.  The term "dynamic unfolding" means that we unfold the forward dynamics model for multiple times, and the actual steps of the unfolding are done dynamically. Namely, we do the unfolding of the dynamics model "on the fly". Roughly speaking, we try to find the best number of timesteps for unfolding by runtime evaluation.

Reviewer1:  Thank you. We will address your problem in the next version.

Reviewer2: Thank you. We will address your problem in the next version.

---

### Decision · Program_Chairs · 2018-01-29
**ICLR 2018 Conference Acceptance Decision**

**Decision:**

Reject

**Comment:**

The paper has some potentially interesting ideas but it feels very preliminary. The experimental section in particular needs a lot more work.